# The Use of Carbon Dioxide as a Green Approach to Recover Bioactive Compounds from Spent Coffee Grounds

**DOI:** 10.3390/foods12101958

**Published:** 2023-05-11

**Authors:** Raffaele Romano, Lucia De Luca, Giulia Basile, Chiara Nitride, Fabiana Pizzolongo, Paolo Masi

**Affiliations:** 1Department of Agricultural Sciences, University of Naples Federico II, Via Università, 100, 80055 Portici, NA, Italy; rafroman@unina.it (R.R.); lucia.deluca@unina.it (L.D.L.); giuli.basile@studenti.unina.it (G.B.); chiara.nitride@unina.it (C.N.); 2CAISIAL—Center of Food Innovation and Development in the Food Industry, University of Naples Federico II, Via Università, 133, 80055 Portici, NA, Italy; masi@unina.it

**Keywords:** antioxidant activity, coffee, polyphenols, caffeine, volatile organic compounds, fatty acids

## Abstract

Spent coffee grounds (SCG) contain bioactive compounds. In this work, given the increasing demand to valorize waste and use green technologies, SCG were submitted to extraction by carbon dioxide (CO_2_) in supercritical and liquid conditions. The extraction parameters were varied to obtain the maximum yield with the maximum antioxidant activity. The use of supercritical and liquid CO_2_ with 5% ethanol for 1 h provided yields (15 and 16%, respectively) comparable to those obtained by control methods for 5 h and extracts with high total polyphenolic contents (970 and 857 mg GAE/100 g oil, respectively). It also provided extracts with DPPH (3089 and 3136 μmol TE/100 g oil, respectively) and FRAP (4383 and 4324 μmol TE/100 g oil, respectively) antioxidant activity levels higher than those of hexane extracts (372 and 2758 μmol TE/100 g oil, respectively) and comparable to those of ethanol (3492 and 4408 μmol TE/100 g oil, respectively). The SCG extracts exhibited linoleic, palmitic, oleic, and stearic acids (predominant fatty acids) and furans and phenols (predominant volatile organic compounds). They were also characterized by caffeine and individual phenolic acids (chlorogenic, caffeic, ferulic, and 3,4-dihydroxybenzoic acids) with well-known antioxidant and antimicrobial properties; therefore, they could be used in the cosmetic, pharmaceutical, and food sectors.

## 1. Introduction

Spent coffee grounds (SCG) are residues obtained from coffee prepared with hot water or steam [1]. Every year, the coffee sector produces approximately 6 million tons of SCG [2]. Recently, food waste recovery has become very important for environmental pollution management, and SCG represent a great pollution hazard if discharged into the environment. Although they contain many valuable substances such as organic carbon, direct SCG when introduced into the soil can be toxic to microorganisms and plants [3]. The use of food waste as an ingredient for the formulation of healthy products could generate economic gains for the industry, produce health benefits, and reduce the environmental impact [4]. Currently, there are various strategies for reusing SCG as biofuel [5], in compost facilities [6], and in the cosmetic industry [7]. Although the World Health Organization considers coffee to be a “non-nutritive dietary component”, it contains numerous bioactive compounds [8]. SCG can be a source of dietary fiber linked to antioxidant compounds, which are useful in the development of new foods with potential health benefits. Fiber from coffee grounds can exert positive control over sugar metabolism, making foods more suitable for diabetics due to the release of bioactive compounds during gastrointestinal digestion. In addition, replacing refined flour with SCGs may contribute to reducing the risk of diseases such as diabetes [9]. Vegetable by-products are a good source of primary and secondary metabolites, such as terpenes, alkaloids, and phenolic compounds; in addition, polyphenols have attracted attention due to their activity against disease [10]. SCG are a source of bioactive compounds such as polyphenols, and coffee polyphenols have many health-promoting properties, such as antioxidant [11], anti-inflammatory, anticancer, antidiabetic, and antihypertensive properties [12]. The predominant phenolic acids found in SCG are caffeic acid, ferulic acid, and chlorogenic acid, as described by Vu et al. [13]. Caffeic and chlorogenic acids have antitumor properties in human cells [14]. The oil extracted from coffee contains excellent percentages (45.04–40.67%) of linoleic acid (C18:2), which is an essential fatty acid that the human body cannot synthesize [15]. These bioactive compounds can be recovered using green technologies in place of conventional solid–liquid extraction based on pollutant solvents to improve process selectivity and safety and to reduce the impact on the environment and the extraction time.

Among the green technologies, supercritical fluid extraction (SFE) using carbon dioxide (CO_2_) is suitable for thermally labile substances, is very safe to use, and allows the elimination of hazardous substances [16]. CO_2_ is a green solvent that is easy to remove from products and is a nonozone-depleting, nontoxic, and nonflammable solvent [17]. CO_2_ was used to recover oil or antioxidant phenolic compounds from SCG [18] and other matrices, such as basil leaves [19], citrus peels [20], and walnut husks [21]. There are various parameters that can potentially be changed, such as the temperature and pressure, which can affect the solubility of various compounds [22].

In this work, SCG from espresso coffee produced by bars were subjected to extraction with CO_2_ in liquid and supercritical conditions to recover bioactive compounds. To our knowledge, no study has examined the possibility to perform the CO_2_ liquid extraction from espresso SCG, and there are few studies in the literature regarding the volatile organic compounds (VOCs) of the extracted oil. The extractions conducted with and without the addition of cosolvent (5% ethanol) were compared to the solvent (100% ethanol and 100% hexane) extractions used as controls. The extracts obtained were characterized for their extraction yield, total polyphenolic content, individual polyphenolic profile, caffeine content, fatty acid profile, antioxidant activity, and VOCs.

## 2. Materials and Methods

### 2.1. Samples

The SCG samples were residues from espresso coffee prepared at bars using a blend of arabica and robusta varieties in the ratio of 85:15 (Toraldo, Caserta, Italy). The espresso coffees were prepared by using a 100S La San Marco industrial coffee machine (Gorizia, Italy). The SCG were frozen at −20 °C, lyophilized for 48 h (at −50 °C, <0.05 mbar), and kept in a dark and dry place.

### 2.2. Chemicals

The carbon dioxide (CO_2_) (assay purity 99.9%) used was purchased from SOL Spa (Naples, Italy). All solvents and reagents used for the experiments were purchased from Sigma–Aldrich Co. (Milano, Italy).

### 2.3. Moisture Content

The moisture content of the SCG was determined by drying the samples for 24 h at 105 °C. The results are expressed as a weight/weight percentages of water (% *w*/*w*).

### 2.4. Extraction Methods

Soxhlet extractions were used as a control method to compare the CO_2_ extraction. Approximately 5 g of dried SCG was weighed, placed in a cellulose extraction thimble (Whatman^®^ 2800432), and plugged with cotton. The thimble was placed inside a Soxhlet extractor, and the extractions were conducted with n-hexane and ethanol separately. The content of the flask (solvent + extracted compounds) was dried at 45 °C using a Rotavapor Labourota4000-Efficient instrument (Heidolph Instrument, Schwabach, Germany) and stored at −18 °C until the analysis. The n-hexane extraction method was coded Cs, and the ethanol method was coded Ce. Both methods lasted 5 h.

In supercritical and liquid CO_2_ extractions, approximately 14 g of dried SCG was added to an SFC 4000 extractor (JASCO International Co., Ltd., Tokyo, Japan) equipped with a 50 mL volume extractor vessel. The CO_2_ extraction was performed for 1 h at a flow rate of 10 mL min^−1^, a pressure rate of 30 MPa, and a temperature of 60 °C for supercritical extraction and 20 °C for liquid extraction. The CO_2_ extractions were performed with 0 and 5% ethanol concentrations as the cosolvents. The extractions by supercritical conditions with 0 and 5% ethanol concentrations were coded SC0 and SC5, respectively. The extractions by liquid conditions with 0 and 5% ethanol concentrations were coded L0 and L5, respectively. The oil extracts were stored at −18 °C until the analysis. All extraction yields are expressed as g extract/100 g dry matter.

### 2.5. Total Polyphenol Content

The total polyphenol content (TPC) of the sample was determined using the Folin–Ciocalteu method reported by Andrade et al. [23] with modifications. To 50 mg of extract, we added 3 mL of methanol, vortexed the mixture for 30 s, and sonicated it for 20 min. Subsequently, the solution was filtered with a 0.22 μm PES filter and to 0.1 mL of this solution 0.5 mL of Folin-Ciocalteu reagent, 7.9 mL of ultrapure H_2_O and 1.5 mL of a 20% Na_2_CO_3_ solution were added. This solution was incubated for 2 h in the dark at room temperature. The absorbance was read at 765 nm using a UV-1601PC UV–visible scanning spectro-photometer (Shimadzu, Milan, Italy). To determine the TPC, a calibration curve in the linearity range of 10–800 mg/L (*R*^2^ = 0.99) was constructed using gallic acid as the standard. The results are reported as mg of gallic acid equivalent (GAE)/100 g of oil.

### 2.6. Individual Polyphenols and Caffeine Assessed by Ultrahigh-Performance Liquid Chromatography Analysis

The individual polyphenol and caffeine contents were determined by an ultrahigh-performance liquid chromatography (UHPLC) analysis according to Romano et al. [20] with some modifications. Five hundred milligrams of extract was weighed and dissolved in 0.5 mL of methanol. The mixture was vortexed for 30 s, sonicated for 20 min, and filtered with a 0.22-μm PES filter (Phenomenex, Torrance, CA, USA), then 10 μL was injected into the UHPLC system. A UHPLC system (Jasco LC-4000 Series, Tokyo, Japan) equipped with a PDA detector MD-4010, Column Oven CO-4061, UHPLC Semimicro Pump PU-4285, and a Spherisorb ODS2 (5 μm, 4.6 mm × 250 mm) C18 reversed-phase column (Waters, Arnhem, The Netherlands) was used. Caffeine detection was performed at 280 nm; chlorogenic acid, caffeic acid, and ferulic acid detection was performed at 330 nm; and 3,4 dihydroxy benzoic acid detection was performed at 260 nm using a photo diode array detector. The identification and quantification of the compounds was performed by comparing the retention times and areas between external standard peaks (dihydroxybenzoic acid, caffeic acid, chlorogenic acid, ferulic acid, and caffeine) and the peaks of the samples. The results are expressed as mg of individual compound/100 g of oil.

### 2.7. Fatty Acid Profile

The fatty acid profile was determined through the use of a gas chromatograph coupled with an FID detector (Agilent Technologies 6850 Series II, Santa Clara, CA, USA) after trans-esterification with 2 N KOH in anhydrous methanol as described by Romano et al. [21]. The GC was equipped with a 90% biscyanopropyl−10% cyanopropyl phenyl siloxane capillary column (100 m, 0.25 mm, ID 0.20 µm) (Supelco, Bellefonte, PA, USA). The oven temperature schedule was 140 °C × 5 min, an increase of 4 °C/min to 175 °C for 20 min, and finally 3 °C/min 240 °C held for 10 min. The injector program temperature was 120 °C × 0.1 min, with an increase of 500 °C/min until reaching 260 °C held for 5 min. Helium was used as the carrier gas (flow rate 2 mL/min). The chromatogram peaks were identified using an external 37-component standard (Supelco, Bellefonte, PA, USA) by comparing the retention times of the standards with those of the samples under the same operating conditions. The results are expressed as relative % values.

### 2.8. Antioxidant Activity Assay

The antioxidant activity of the extracts was determined by DPPH^●+^, ABTS^●+^, and FRAP assays.

The 2,2-diphenyl-1-picrylhydrazyl (DPPH^●+^) assay was performed as described by Barajas-Álvarez et al. [24] with modifications. Three milliliters of methanol was added to 50 mg of extract, and the mixture was vortexed for 30 s, sonicated for 20 min, and filtered with a 0.22-µm PES filter. Subsequently, to 0.2 mL of this solution we added 2 mL of 0.5 mM methanolic DPPH solution. The mixture was kept in the dark for 30 mins and the absorbance was measured at 516 nm using a UV-1601PC UV–visible scanning spectrophotometer (Shimadzu, Milan, Italy). To determine the antioxidant activity in the extracts, a calibration curve with Trolox as the standard was prepared in the linearity range of 10–200 μM (*R*^2^ = 0.99). The results are expressed as μmol Trolox equivalent (TE)/100 g oil.

The 2,2′-azinobis-3-ethyl-benzothiazoline-6-sulfonic acid (ABTS^●+^) assay was performed as described by Romano et al. [20], with modifications. Three milliliters of methanol was added to 50 mg of extract, and the mixture was vortexed for 30 s, sonicated for 20 min, and filtered with a 0.22-µm PES filter. A quantity of 400 μL of diluted extract was mixed with 3 mL of ABTS^●+^ working solution. The solution was kept in the dark for 5 min and the absorbance was measured at 751 nm using a spectrophotometer. The antiradical activity was calculated using a Trolox calibration curve in the linearity range of 10–200 μM (*R*^2^ = 0.99). The results are expressed as μmol Trolox equivalent (TE)/100 g oil.

The ferric reducing antioxidant power (FRAP) assay was performed following Romano et al. [19] with some modifications. The FRAP assay utilizes antioxidants as reductants in a redox colorimetric reaction. At low pH, the rapid reduction of the ferric tripyridyl triazine (Fe III TPTZ) complex to ferrous tripyridyl triazine (Fe II-TPTZ) by antioxidants was detected at 593 nm. Three milliliters of methanol was added to 50 mg of extract, and the mixture was vortexed for 30 s, sonicated for 20 min, and filtered with a 0.22-µm PES filter. An aliquot of 70 μL of diluted extract was mixed with 279 μL of FRAP reagent and 651 μL of acetate buffer. After 5 min of incubation at ambient temperature in the dark, the absorbance was recorded at 593 nm. The antioxidant activity was calculated using a Trolox calibration curve in linearity range of 10–200 μM (*R*^2^ = 0.99). The results are expressed as μmol Trolox equivalent (TE)/100 g oil.

### 2.9. Volatile Organic Compounds

The analysis of volatile organic compounds (VOCs) in obtained extracts was performed using the solid-phase microextraction technique (SPME) followed by gas chromatography coupled to mass spectrometry (GC/MS) according to Gomes et al. [25] with modifications. Briefly, 150 mg of extract was added to a 10 mL vial, and the sample was kept at 60 °C for 10 min. Subsequently, a divinylbenzene/carboxen/polydimethylsiloxane (DVB/CAR/PDMS) fiber (Supelco, Bellefonte, PA, USA) was inserted into the vial and maintained at 60 °C for 10 min. Successively, the fiber was introduced into the gas chromatograph and the thermal desorption of the analytes was performed at 250 °C for 7 min in splitless mode. The GC system equipped with a mass detector was purchased from Agilent (Santa Clara, CA, USA). The VOC separation was performed on an HP-5MS 5% diphenyl/95% dimethylpolysiloxane capillary column (30 m × 0.25 mm ID × 0.25 µm) (Agilent). The oven temperature was set to 40 °C for 2 min, then subsequently it was increased from 40 °C to 160 °C at 6 °C min^−1^ and from 160 to 210 °C at 10 °C min^−1^. The gas carrier helium was set to 1 mL/min. The ionizing electron energy was 70 eV, and the mass-to-charge ratios were scanned over the range of 40 to 450 amu in full-scan acquisition mode. The temperatures of the injection and ion source were 250 and 230 °C, respectively. The compounds were identified using the NIST (National Institute of Standards and Technology) Atomic Spectra Database version 2.0 and verified for retention indices. The result are expressed as relative percentages (%) of VOCs.

### 2.10. Statistical Analysis

All experiments were performed in triplicate, and the results are expressed as the mean values (±standard deviations) of the three replicates. The data were submitted to a one-way analysis of variance (ANOVA) and Tukey’s multiple-range test (*p* ≤ 0.05) using XLSTAT software (Addinsoft, New York, NY, USA).

## 3. Results and Discussion

### 3.1. Extraction Yields

The SCG contained 65.24% *w*/*w* moisture, in agreement with the findings of Juarez et al. [26], who reported a value of 63.43%. In Table 1, the extraction yields for the different methods used are reported. The minimum yield values were recorded in the SC0 and L0 extractions (14.21 and 14.48, respectively) and the maximum in the controls Ce and Cs (15.95% and 16.35%, respectively). The SC0 yield was similar to the yield found by Couto et al. [27], who reported a value of 14.7% of oil from spent coffee extracted at 30 MPa and 50 °C for 3 h. The addition of ethanol in SC5 and L5 increased the yield, reaching values of 15.45 and 15.88%, respectively, which were not significantly different from the controls Ce and Cs. Additionally, Ahangari and Sargolzaei [28] found that increases in the volume of modifier resulted in an increase in extraction yield. Coelho et al. [29] showed that at 20 MPa and 313 K, the increase in the concentration of ethanol added within 5–10% increased the yield from 10.7% to 11.4% and significantly decreased the extraction time from 100 to 58 min. The addition of ethanol increased the local density around the molecules, which resulted in increased interactions between the cosolvent and solute molecules, such as H-bonding [29]. Using the SC5 and L5 methods, it is possible to decrease the extraction time from 5 to 1 h without a substantial reduction in yield with respect to the control methods.

### 3.2. Polyphenol and Caffeine Content

The total polyphenolic content (TPC) of the oil extracted from SCG is shown in Table 2. The lowest TPC value was recorded with the Cs method (73.00 mg GAE/100 g oil). The values increased up to 419.50 and 692.75 mg GAE/100 g oil in L0 and SC0, respectively. The use of 5% ethanol led to values of 857.25 mg GAE/100 g oil in SC5 and 969.75 mg GAE/100 g oil in L5. This trend was also observed by Sharma et al. [30], who reported that the value of 37.10 increased to 55.45 mg GAE/100 g oil with the addition of 30% ethanol during extraction. The increase in TPC was proportional to the increase in ethanol due to the polarity of polyphenols. Araújo et al. [31] reported higher values (411 mg GAE/100 g extract) in extracts obtained with supercritical CO_2_ and ethanol in a ratio of 0.5:1 at 15 MPa and 60 °C, but this result was lower than our results because our extractions were conducted at 30 MPa. In fact, Alexandre et al. [32] showed that increasing the pressure from 100 bar to 250 bar at 70 °C resulted in an increase in TPC from 27.7 to 37.0 (mg GAE/g d.w) in strawberry tree extracts. The highest value of TPC was obtained with Ce (1312.00 mg GAE/100 g oil).

The 3,4 dihydroxybenzoic, caffeic, chlorogenic, and ferulic acids were detected in Ce, SC5, and L5 (Table 2) due to the ethanol polarity and consequent affinity for the polyphenols. They were not found in the Cs, L0, and SC0 extracts, in agreement with Araújo et al. [31], who obtained the same results.

Here, 3,4 dihydroxybenzoic acid (protocatechuic acid) was the predominant phenolic acid, in agreement with Araújo et al. [31], in the range of 4.70–9.08 mg/100 g oil, with the highest value in L5 because the liquid extraction with CO_2_, due to the temperature of 20 °C, did not degrade the phenolic compounds [20]. Semaming et al. [33] reported some beneficial functions of this phenolic acid, such as anti-inflammatory, antihyperglycemic, and antiapoptotic effects. Moreover, in vitro studies have shown that protocatechuic acid has antimicrobial properties and exerts synergistic interactions with some antibiotics against pathogens.

Caffeic acid was found in the range of 1.04–3.16 mg/100 g oil, with the highest value in SC5. The addition of 5% ethanol led to a recovery of this compound in both the SC5 and L5 extractions, without statistically significant differences between them. Caffeic acid is a potent antioxidant that presents important anti-inflammatory actions due to the inhibition of lipooxygenase activity [34]. Jung et al. [35] evaluated the antihyperglycemic properties of caffeic acid, and the data obtained suggest that caffeic acid is an effective antidiabetic agent via its ability to enhance insulin secretion and decrease the hepatic glucose output, along with the increased level of adipocyte glucose disposal in type 2 diabetic animals.

Chlorogenic acid was found at a concentration of 2.01 mg/100 g oil in L5 and 1.40 mg/100 g oil in SC5. Grujic et al. [36] evaluated the extraction of chlorogenic acid from mate tea with liquid CO_2_ and ethanol and found a higher concentration of chlorogenic acid in the extract with liquid CO_2_ plus ethanol, which could be explained by the higher selectivity of liquid CO_2_ with ethanol for the chlorogenic acid. It is known that changes in the polarity of the solvent adopted in the extraction alter the solvent power for antioxidant compounds. Chlorogenic acid has many biological activities, such as protection from cardiovascular diseases, antioxidant activity, prevention of septic arthritis caused by *Candida albicans*, modification of plasma and liver concentrations of cholesterol and triacylglycerol, and neuroprotection [37].

SC5 and L5 showed increases in ferulic acid (1.96–1.89 mg/100 g oil, respectively) compared to SC0 and L0, but no statistically significant differences were found. Bitencourt et al. [38] showed the solubility of ferulic acid in systems with pure supercritical CO_2_ and supercritical CO_2_ mixed with ethanol, and it was observed that the addition of 5% cosolvent increased the ferulic acid solubility by up to 30 times compared with the solubility of ferulic acid in pure supercritical CO_2_. The increase in the solubility of ferulic acid during extraction with the addition of ethanol is a consequence of the increase in solvent density caused by the introduction of the cosolvent in the system. However, the most significant increase in solubility is due to hydrogen bonds formed between the solute and ethanol [38].

Caffeine is the most commonly used substance with a centrally excitatory effect, since it is a common ingredient in beverages and foods. According to data published by the European Food Safety Authority (EFSA), the average daily consumption rate of caffeine in young adults (18–65 years old) is 37–319 mg and is derived mainly from products based on coffee and cocoa beans, tea leaves, guarana berries, and kola nuts [39]. Caffeine is an alkaloid semipolar [40], and the addition of polar cosolvents such as ethanol resulted in solubility enhancements of caffeine compared to extractions with pure carbon dioxide. In fact, there was an increase in caffeine from SC0 (230.44 mg/100 g oil) to SC5 (579.21 mg/100 g oil) and from L0 (120.58) to L5 (276.75).

The addition of 5% ethanol to supercritical and liquid CO_2_ led to 49% and 43% increases in the caffeine content and phenolic acids, respectively.

There were differences in the polyphenol and caffeine contents in each study, which may have been due to the differences between the coffee blends, since coffee varieties have different contents of caffeine and polyphenols (arabica or robusta).

### 3.3. Fatty Acid Composition

The most abundant fatty acids found in all extracts were linoleic acid C18:2 (48.97–45.18%), palmitic acid C16:0 (32.21–27.84%), oleic acid C18:1 (10.24–9.58%), and stearic acid C18:0 (8.21–7.47%), as reported in Table 3. The extraction method did not affect the fatty acid profiles; in fact, no statistically significant differences were found among the samples. The high percentage of linoleic acid gives fluidity to the extract and a low melting point. Oils extracted from SCG have a fatty acid profile similar to cottonseed oil, with a high content of polyunsaturated fatty acids (PUFAs), in particular linoleic acid, as reported by Knothe [41]. The values obtained were similar to the values reported by Coelho et al. [29] and Araújo et al. [31] in SCG, which showed concentration ranges of 33.87–33.19% and 33.04–29.70% for palmitic acid, 42.98–42.59% and 45.79–44.05% for linoleic acid, 11.5–11.1% and 8.89–8.43% for oleic acid, and 7.51–7.28% and 8.79–7.69% for stearic acid, respectively.

Myristic acid (C14:0), myristoleic acid (C14:1), heptadecanoic acid (C17:0), cis-11eicosanoic acid (C20:1), eneicosanoic acid (C21:0), cis-11,14 eicosadienoic acid (C20:2), tricosanoic acid (C23:0), and lignoceric acid (C24:0) were found in concentrations <1%. The fatty acid composition of coffee oil determines its emollient property, which has the capacity to block sunlight harmful to human skin, and this benefit can be exploited in the cosmetics industry [42]. In this way, oil extracted from SCG exerts activity against melanogenesis related to unsaturated fatty acids (linoleic and oleic acids) in cell culture and animal models [43].

### 3.4. Antioxidant Activity

Different in vitro antioxidant assays (DPPH^●+^, FRAP, and ABTS^●+^) were performed to take into account the various modes of action of antiradicals (Table 4). The highest antioxidant activity by DPPH^●+^ and FRAP assays was recorded in the SC5 (3089 and 4383 µmolTE/100 g oil) and L5 samples (3136 and 4324 µmolTE/100 g oil), and was not significantly different from the control Ce. The SC0 and L0 extracts and Cs control showed the lowest antioxidant activity. This was due to the nonpolarity of CO_2_ and hexane, which cannot extract polar polyphenols with antioxidant properties. The use of ethanol increased the antioxidant activities, and the Ce control was shown to be the highest by all three assays. A positive linear relationship was found between the antioxidant activity and TPC. This correlation was also found and described by Muflihah et al. [44], who reported that the phenolic acids contributed most.

The ABTS·+ antiradical activity showed high values (5508, 5396, and 8351 µmolTE/100 g, respectively) in SC5, L5, and Ce, which corresponded to the extracts with the highest TPC. The values obtained by the DPPH assay were similar to the results obtained by Araújo et al. [31], and the ABTS results followed the same trend. The addition of 5% ethanol in supercritical CO_2_ provided increases of 647% in DPPH antioxidant activity, 29% in FRAP, and 81% in ABTS. No statistically significant difference in antioxidant activity measured by DPPH and FRAP assays was found between SC5, L5, and Ce. Furthermore, the addition of 5% cosolvent in liquid CO_2_ produced increases of 746% in the DPPH antioxidant activity and 69% in FRAP, and a decrease of 4% in ABTS. The decrease in ABTS activity was presumably due to the contribution of lipophilic antioxidants such as *α*-tocopherol, *β*-tocopherol, campesterol, stigmasterol, and sitosterol [45], which are extracted in higher quantities by liquid CO_2_ than by supercritical CO_2_; in addition, the ABTS radical used in the assay is related to hydrophilic and lipophilic compounds, while DPPH is related to hydrophilic compounds [46].

### 3.5. Volatile Organic Compounds

The volatilome of coffee is very complex (including sulfur compounds, pyrazines, pyridines, pyrroles, furans, aldehydes, alcohols, ketones, esters, and phenols). The identified VOCs are mostly generated by thermal reactions during the roasting and brewing of coffee [47]. There are few studies in the literature on the VOC compositions of supercritical CO_2_ extracts from SCG, among which de Toledo et al. [48] found that the addition of ethanol has a positive influence on the removal of polar volatile compounds [49].

The most abundant VOCs identified belong to the category of furans and phenols (Table 5).

Furans originate during the roasting process of coffee beans by the Maillard reaction (MR) [50]. Among furans, 2-furanmethanol was the most abundant compound identified with the highest contents in the SC5 and L0 extracts (54.46% and 51.12%, respectively), while a low percentage of this compound was found in the Cs extract (1.91%).

Lactones (γ-butyrolactone) and phenols are the degradation products of chlorogenic acid, and their production is correlated with roasting. In general, γ-butyrolactone and its alkyl-substituted derivatives give butter or coconut flavors and have been used for food flavorings [51]. Moon and Shibamoto [51] reported that 2-methoxyphenol and 2-methoxy-4-vinylphenol decreased with the increasing roasting intensity in coffee seeds, but there was also an increase in phenol.

Pyrroles are not very responsible for the aroma of coffee but may contribute to the sweet and slightly burnt taste [48].

Among pyrazines, 2-ethyl-6-methyl-pyrazine and 2-ethyl-3,5-dimethyl-pyrazine were detected. The pyrazines contribute to the nutty, earthy, roasted, and green coffee flavors [52].

Hurtado-Benavides et al. [53] reported the flavor descriptors of some molecules found in the extracts of this study: furanmethanol (caramel, hot oil, bitter), 2-metoxy-4-vinylphenol (apple, hot spicy, peanut, wine clove, curry), 4-ethyl-2-methoxyphenol (spicy), and furfuryl acetate (roasted nut, floral, and alcohol sensations).

## 4. Conclusions

Espresso SCG are not just a waste product but an excellent source of antioxidant compounds that can be recovered through green technology based on the use of CO_2_. Liquid and supercritical CO_2_ extractions with the use of 5% ethanol showed yields (16 and 15%, respectively) similar to those obtained with control methods, where ethanol and hexane were used. They allowed time savings because they lasted 1 h versus 5 h and provided oils containing high contents of total phenolic compounds (970 and 857 mg GAE/100 g oil, respectively) and furans and phenols as the predominant volatile organic compounds.

Moreover, they reduced the environmental pollution because CO_2_ is a nontoxic, nonflammable, recyclable, and “green” solvent. Therefore, extraction with a CO_2_ and ethanol mixture could be a valid alternative to traditional solvent extraction for waste recovery. From a circular economy perspective, coffee extract could have many applications, such as its use as a fat source in foods, for the development of active packaging for coffee to prevent lipid oxidation, or the possibility of using coffee oil as a flavoring ingredient in foods.

## Figures and Tables

**Table 1 foods-12-01958-t001:** Yields (%) of oils extracted from spent coffee grounds by different extraction methods (SC0, SC5, L0, and L5) and control methods (Ce and Cs).

Code	Extraction Methods	Time (h)	% Ethanol (Cosolvent)	Yield (g/100 g d.m.)
Ce	Ethanol	5	0	15.95 ± 0.11 ^a^
Cs	n-Hexane	5	0	16.35 ± 1.71 ^a^
SC0	Supercritical CO_2_	1	0	14.21 ± 0.15 ^b^
SC5	Supercritical CO_2_	1	5	15.45 ± 0.09 ^ab^
L0	Liquid CO_2_	1	0	14.48 ± 0.07 ^b^
L5	Liquid CO_2_	1	5	15.88 ± 0.18 ^a^

^a–b^ Different letters in the same row indicate statistically significant differences (*p* < 0.05).

**Table 2 foods-12-01958-t002:** Total phenolic compounds (TPCs), individual phenolic acids, and caffeine contents in oils extracted from spent coffee grounds by different extraction methods (SC0, SC5, L0, and L5) and control methods (Ce and Cs).

	Ce	Cs	SC0	SC5	L0	L5
TPC (mgGAE/100 g oil)	1312.00 ± 19.00 ^a^	73.00 ± 1.53 ^d^	692.75 ± 55.00 ^bc^	857.25 ± 37.00 ^b^	419.50 ± 66.00 ^c^	969.75 ± 35.00 ^b^
Phenolic acids (mg/100 g oil):						
3,4-Dihydroxybenzoic acid	5.08 ± 0.46 ^b^	n.d.	n.d.	4.70 ± 0.11 ^b^	n.d.	9.08 ± 0.39 ^a^
Caffeic acid	1.04 ± 0.05 ^b^	n.d.	n.d.	3.16 ± 0.04 ^a^	n.d.	3.04 ± 0.87 ^a^
Chlorogenic acid	2.62 ± 0.01 ^a^	n.d.	n.d.	1.41 ± 0.16 ^c^	n.d.	2.01 ± 0.06 ^b^
Ferulic acid	2.47 ± 0.02 ^a^	n.d.	n.d.	1.96 ± 0.03 ^b^	n.d.	1.89 ± 0.12 ^b^
Caffeine	378.87 ± 30.98 ^b^	15.07 ± 0.77 ^e^	230.44 ± 31.23 ^cd^	579.21 ± 59.15 ^a^	120.58 ± 1.82 ^de^	276.75 ± 2.35 ^bc^

^a–e^ Different letters in the same row indicate statistically significant differences (*p* < 0.05). Note: n.d.: not detected.

**Table 3 foods-12-01958-t003:** Relative percentage (%) values of the most abundant (>1%) fatty acids in oil extracted from spent coffee grounds by different extraction methods (SC0, SC5, L0, and L5) and control methods (Ce and Cs).

Fatty Acid (%)	Ce	Cs	SC0	SC5	L0	L5
Palmitic acid (C16:0)	32.21 ± 0.51	30.51 ± 0.19	31.38 ± 0.06	27.84 ± 3.52	30.52 ± 0.35	31.06 ± 1.18
Stearic acid (C18:0)	7.47 ± 0.05	7.77 ± 0.08	7.47 ± 0.09	8.21 ± 0.55	7.65 ± 0.08	7.96 ± 0.29
Oleic acid (C18:1)	9.70 ± 0.23	10.17 ± 0.29	9.58 ± 0.28	10.24 ± 0.41	10.09 ± 0.24	10.22 ± 0.01
Linoleic acid (C18:2)	46.14 ± 0.07	46.01 ± 0.21	46.36 ± 0.06	48.97 ± 3.68	46.17 ± 0.25	45.18 ± 0.87
Arachidic acid (C20:0)	2.08 ± 0.34	2.48 ± 0.10	2.35 ± 0.17	2.01 ± 0.64	2.42 ± 0.13	2.51 ± 0.13
Linolenic acid (C18:3)	1.21 ± 0.13	1.30 ± 0.03	1.23 ± 0.05	1.16 ± 0.18	1.32 ± 0.04	1.23 ± 0.19
Cis-8,11,14 Eicosatrienoic acid (C20:3)	0.75 ± 0.24	1.09 ± 0.04	1.03 ± 0.13	0.99 ± 0.20	1.05 ± 0.06	1.18 ± 0.08
Σ SFA	41.99 ± 0.09	41.11 ± 0.01	41.54 ± 0.18	38.36 ± 3.63	41.01 ± 0.15	41.91 ± 1.33
Σ MUFAs	9.90 ± 0.21	10.46 ± 0.29	9.82 ± 0.32	10.47 ± 0.34	10.41 ± 0.30	10.45 ± 0.17
Σ PUFAs	48.11 ± 0.30	48.43 ± 0.31	48.64 ± 0.14	51.17 ± 3.28	48.58 ± 0.15	47.64 ± 1.17
ω-6/ω-3	39.00 ± 4.19	36.24 ± 0.89	38.63 ± 2.04	44.11 ± 10.58	35.60 ± 1.12	38.28 ± 5.12

No statistically significant differences (*p* < 0.05) were found.

**Table 4 foods-12-01958-t004:** Antioxidant activity (µmolTE/100 g oil) values measured by DPPH, FRAP, and ABTS in oils extracted from spent coffee grounds by different extraction methods (SC0, SC5, L0, and L5) and control methods (Ce and Cs).

	Ce	Cs	SC0	SC5	L0	L5
DPPH	3492.30 ± 28.10 ^a^	371.85 ± 29.43 ^b^	413.40 ± 70.52 ^b^	3088.50 ± 11.55 ^a^	370.50 ± 47.81 ^b^	3135.90 ± 54.71 ^a^
FRAP	4408.10 ± 14.23 ^a^	2758.10 ± 21.09 ^c^	3400.80 ± 21.45 ^b^	4382.60 ± 55.34 ^a^	2557.30 ± 64.22 ^c^	4323.60 ± 38.72 ^a^
ABTS	8351.00 ± 14.35 ^a^	890.99 ± 89.10 ^d^	3048.10 ± 15.32 ^c^	5508.30 ± 11.22 ^b^	5648.00 ± 11.04 ^b^	5396.30 ± 30.05 ^b^

^a–d^ Different letters in the same row indicate statistically significant differences (*p* < 0.05).

**Table 5 foods-12-01958-t005:** Relative percentage (%) values of volatile organic compounds in oils extracted from spent coffee grounds by different extraction methods (SC0, SC5, L0, and L5) and control methods (Ce and Cs).

	Ce	Cs	SC0	SC5	L0	L5
Ʃ Furans	60.08 ^b^	25.12 ^e^	37.50 ^d^	66.85 ^a^	61.11 ^ab^	45.15 ^c^
Furfurale	1.00 ± 0.13 ^d^	9.72 ± 0.63 ^a^	n.d.	4.32 ± 0.13 ^c^	n.d.	6.93 ± 0.35 ^b^
2-Furanmethanol	51.01 ± 0.75 ^a^	1.91 ± 0.31 ^c^	30.45 ± 2.11 ^b^	54.46 ± 0.54 ^a^	51.12 ± 0.08 ^a^	33.88 ± 2.36 ^b^
2,2′-methylenebis-furan	2.43 ± 0.07 ^c^	4.63 ± 0.05 ^a^	2.41 ± 0.08 ^b^	2.43 ± 0.25 ^bc^	3.24 ± 0.29 ^a^	1.40 ± 0.01 ^c^
2-Furanmethanol acetate	3.59 ± 0.16 ^cd^	5.97 ± 0.49 ^ab^	2.39 ± 0.50 ^cd^	4.15 ± 0.25 ^bc^	6.75 ± 0.83 ^a^	2.03 ± 0.08 ^d^
2-(2-furanylmethyl)-5-methyl-furan	2.05 ± 0.15 ^bc^	2.89 0.18 ^a^	2.25 ± 0.12 ^b^	1.49 ± 0.25 ^cd^	n.d.	0.91 ± 0.04 ^d^
Ʃ Lactone	11.07 ^a^	11.94 ^a^	3.32 ^c^	14.39 ^a^	6.93 ^b^	7.05 ^b^
Butyrolactone	11.07 ± 0.19 ^a^	11.94 ± 0.06 ^a^	3.32 ± 0.30 ^c^	14.39 ± 1.94 ^a^	6.93 ± 0.75 ^b^	7.05 ± 0.29 ^b^
Ʃ Phenols	15.17 ^d^	32.58 ^b^	39.22 ^a^	9.03 ^e^	13.38 ^d^	19.56 ^c^
Phenol	2.78 ± 0.32 ^b^	3.75 ± 0.03 ^b^	6.08 ± 0.24 ^a^	1.20 ± 0.03 ^c^	3.95 ± 0.85 ^b^	5.60 ± 0.16 ^a^
2-methoxyphenol	5.24 ± 0.18 ^b^	7.41 ± 0.79 ^a^	8.77 ± 1.06 ^a^	2.87 ± 0.12 ^c^	2.17 ± 0.05 ^c^	3.28 ± 0.16 ^c^
4-ethyl-2-methoxyphenol	6.46 ± 0.01 ^bc^	21.42 ± 1.81 ^a^	19.90 ± 0.39 ^a^	3.90 ± 0.68 ^c^	5.01 ± 0.55 ^c^	9.76 ± 0.04 ^b^
2-methoxy-4-vinylphenol	0.69 ± 0.02 ^cd^	n.d.	4.47 ± 0.48 ^a^	1.06 ± 0.15 ^c^	2.16 ± 0.11 ^b^	0.94 ± 0.12 ^c^
Ʃ Pyrazines	5.66 ^d^	12.20 ^b^	5.94 ^d^	5.09 ^d^	7.59 ^c^	15.43 ^a^
2-ethyl-6-methyl-pyrazine	1.58 ± 0.33 ^d^	8.01 ± 0.56 ^b^	2.76 ± 0.11 ^c^	2.27 ± 0.22 ^cd^	2.40 ± 0.12 ^cd^	13.18 ± 1.23 ^a^
2-ethyl-3,5-dimethyl-pyrazine	4.08 ± 0.16 ^ab^	4.19 ± 0.72 ^a^	3.18 ± 0.02 ^bc^	2.82 ± 0.32 ^bc^	5.19 ± 0.38 ^a^	2.24 ± 0.08 ^c^
Ʃ Pyrroles	8.04 ^d^	18.16 ^a^	14.03 ^b^	4.66 ^e^	11.08 ^c^	12.81 ^bc^
1H-pyrrole-2-carboxaldehyde	0.97 ± 0.01 ^de^	2.49 ± 0.12 ^bc^	2.87 ± 0.38 ^b^	0.71 ± 0.13 ^e^	5.05 ± 0.32 ^a^	1.85 ± 0.28 ^cd^
5-methyl-1H-pyrrole-2-carboxaldehyde	1.44 ± 0.01 ^a^	n.d.	n.d.	n.d.	n.d.	n.d.
1-(1H-pyrrol-2-yl)-ethanone	2.57 ± 0.1 ^e^	12.68 ± 0.38 ^a^	6.22 ± 0.96 ^c^	1.58 ± 0.44 ^e^	4.10 ± 0.40 ^d^	9.10 ± 0.20 ^b^
1-(2-furanylmethyl)1H-pyrrole	3.06 ± 0.21 ^ab^	2.99 ± 0.44 ^ab^	4.94 ± 0.81 ^a^	2.37 ± 0.31 ^b^	1.93 ± 0.40 ^b^	1.86 ± 0.36 ^b^

^a–e^ Different letters in the same row indicate statistically significant differences (*p* < 0.05). Note: n.d.: not detected.

## Data Availability

Data is contained within the article.

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
