# Peer review of "The Use of Carbon Dioxide as a Green Approach to Recover Bioactive Compounds from Spent Coffee Grounds"

_foods, 2023, doi:10.3390/foods12101958_

Round 1

Reviewer 1 Report

Hi dear Editorial board and the respected authors

This article " The use of carbon dioxide as a green approach to recover bioactive compounds from spent coffee grounds” was revised and has a novelty and I recommend the following comments.

·        Please include the background of your study specially for applying the spent coffee grounds.

·        Line 10-11: The type of statistical design used in this research should be mentioned. As well as the dependent and independent variables would be mentioned.

·        Line 16 etc:In the throught the manuscript text dont include standard error or standard deviation.

·        Line 15-17: Please include detail of your data for FRAP and DPPH methods. As well as for hexane and ethanol solvent.

Introduction:

·         Use food waste as an enrichment in food formulation as an example in the introduction. For example, you can refer to the following articles: https://doi.org/10.3390/foods11203263

Materials:

·                      Please write materials as Company Name (City, Country), especially for chemical analysis assessment which used in the study.

Methodology:

·        Line 142: “The results are expressed as μmol Trolox equivalent (TE)/100 g oil.  And Line 161-162: “ why did you assay based on TE/100 g oil it was better express based on TE/ 100 g coffee ground. Please recheck it .

·        Why do you think that bioactive compounds are only in the oily phase and are extracted from the oily phase only by organic solvents? For example in line 209 etc,.

Results:

·        Table 3: Please have a statistical assessment for data.

·        Table 4: Averages and standard deviations should be equalized in terms of decimals.

Discussion:  Discussion text must grammar improve and in some cases it is very weak and maybe there is no discussion at all.

The article has many flaws in express and concept of English, it is suggested to be revised in a scientific and native way.

The article has many flaws in express and concept of English, it is suggested to be revised in a scientific and native way.

Reviewer 2 Report

The paper:  “The use of carbon dioxide as a green approach to recover bioactive compounds from spent coffee grounds”, presents good work on that subject.

 The abstract reflects the work, and the short introduction is enough to introduce the subject.

 One important question that stays after reading the paper is:

1.      If the SCGs are lyophilized, why is necessary to mention or refer to the moisture content? What is the importance of that?

2.      Table 1- the total yield with ethanol is lower than with hexane. Why.? Usually, the yield is higher with a polar solvent like ethanol than with hexane. Any explanation?

3.      How the authors chose the pressure and temperatures to work?

4.      References 15, 25, 26, 28, 29 and others must be revised. Some are not complete at all, and others contain mistakes in the words. Please revise all.

Table 5 is an improvement of the paper when compared with other similar works.

Reviewer 3 Report

This study presents evaluation of extracting polyphenols and bioactives from SCG. The work is relevant to the field given the increasing need to find alternatives to landfill, AD and biomass burning to deal with increasing amounts of food waste. The work is well presented, the methods are clear, explanations for the observations are scientifically sound and backed up by other references.

However, there are two main issues potentially undermining the relevance of the manuscript which should be addressed.

1.       Given that for the majority of results the authors cite their similarity to other published work, it is not clear what they have actually done that is new/original/has not already been published. Please clarify the novelty that this study brings to the field.

2.       Given that the control with ethanol provides highest yields and concentration of polyphenols, is the SFE actually needed at all? The authors have pointed out that CO2 is a green solvent – however, so is ethanol – and that the extraction time is cut from 5 hours to 1. However, does this time cut compensate for the increased energy needed to pressurise the CO2 to supercritical or liquid phase, as well as the other energy costs typically needed to operate an SFE rig (Back Pressure Regulator, pump etc). There needs to be a much higher level critique of the pros and cons of each system to be able to actually justify this as a greener method – including the criteria by which they are judging something to be green to begin with.

Other minor points to address:

Line 31 - ‘SCG represent  great pollution hazard if discharged into the environment’. Need to say a few lines what the pollution hazards associated with SCG are and provide citations.

Line 35 – ‘These residues are expected to have properties similar to those of coffee beans and could therefore be exploited for different industrial applications.’ The citation provided here compares SCG to coffee silverskins, not beans. It’s also unclear as to whether the beans referred to here are the beans before, or after the roasting process. Either way, this statement is highly questionable since the roasting process induces many chemical reactions within the bean and after the process, many of the water soluble chemicals are washed out in the process of making the drink.

Line 63 – ‘The SCG samples were residues from instant coffee preparation performed in bars using a blend of Arabica and Robusta varieties in the ratio of 85:15’. I think the term ‘instant coffee’ is being mistakenly used here. Instant coffee is typically referring to freeze-dried coffee that is then redissolved in water to make the drink. As such, no SCG residues should be being produced at the bars, they should all be at the factory. Do the authors perhaps mean filter coffee (which IS typically made at bars, restaurants etc and WILL produce SCGs)? Please clarify.

Table 2 – TPC content is given twice.

Just a couple of possible translation errors that I have already asked for clarity on in the comments. Other than that, the quality of English is strong throughout.
